# Predicting Frailty and Geriatric Interventions in Older Cancer Patients: Performance of Two Screening Tools for Seven Frailty Definitions—ELCAPA Cohort

**DOI:** 10.3390/cancers14010244

**Published:** 2022-01-04

**Authors:** Claudia Martinez-Tapia, Marie Laurent, Elena Paillaud, Philippe Caillet, Emilie Ferrat, Jean-Léon Lagrange, Jean-Paul Rwabihama, Mylène Allain, Anne Chahwakilian, Pascaline Boudou-Rouquette, Sylvie Bastuji-Garin, Etienne Audureau

**Affiliations:** 1Université Paris Est Créteil (UPEC), INSERM, IMRB, F-94010 Creteil, France; claudia.tapia@aphp.fr (C.M.-T.); marie.laurent@aphp.fr (M.L.); elena.paillaud@aphp.fr (E.P.); philippe.caillet@aphp.fr (P.C.); emilie_frisouille@yahoo.fr (E.F.); jean-paul.rwabihama@aphp.fr (J.-P.R.); mylene.allain@aphp.fr (M.A.); sylvie.bastuji-garin@aphp.fr (S.B.-G.); 2Internal Medicine and Geriatric Department, AP-HP, Hôpital Henri-Mondor, F-94010 Creteil, France; 3Geriatric Oncology Unit, AP-HP, Hôpital Europeen Georges Pompidou, F-75015 Paris, France; 4Primary Care Department, School of Medicine, Université Paris Est Créteil (UPEC), F-94010 Créteil, France; 5Department of Medical Oncology, AP-HP, Hôpital Henri-Mondor, F-94010 Creteil, France; jean-leon.lagrange@aphp.fr; 6Geriatric Department, AP-HP, Hôpital Joffre-Dupuytren, F-91210 Draveil, France; 7Clinical Research Unit (URC Mondor), AP-HP, Hôpital Henri-Mondor, F-94010 Creteil, France; 8Oncogeriatrics, Geriatric Department, AP-HP, Hôpital Broca, F-75013 Paris, France; anne.chahwakilian@aphp.fr; 9Department of Medical Oncology, AP-HP, Hôpital Cochin, F-75014 Paris, France; pascaline.boudou@aphp.fr; 10Public Health Department, AP-HP, Hôpital Henri-Mondor, F-94010 Creteil, France

**Keywords:** geriatric assessment, reference standard, frailty, surveys and questionnaires, sensitivity and specificity

## Abstract

**Simple Summary:**

Screening tools have been developed to identify patients warranting complete geriatric assessment (GA). However, GA lacks standardization and does not capture important aspects of geriatric oncology practice such as actual treatment decisions based on GA findings, expert-based clinical classifications, and/or broader approaches to frailty. We compared the diagnostic performance of screening tools G8 and modified G8 according to: (1) the detection of ≥1 or (2) ≥2 GA impairments, (3) the prescription of ≥1 geriatric intervention and identification of an unfit profile according to (4) a latent class typology, expert-based classifications from (5) Balducci, (6) the International Society of Geriatric Oncology task force (SIOG), and (7) a GA frailty index according to the Rockwood accumulation of deficits. Our findings support the clinical value of the original and modified G8 for detecting a variety of health profiles evocative of frailty in older cancer patients, with evidence of better diagnostic performance of the modified G8 than that of the original G8.

**Abstract:**

Screening tools have been developed to identify patients warranting a complete geriatric assessment (GA). However, GA lacks standardization and does not capture important aspects of geriatric oncology practice. We measured and compared the diagnostic performance of screening tools G8 and modified G8 according to multiple clinically relevant reference standards. We included 1136 cancer patients ≥ 70 years old referred for GA (ELCAPA cohort; median age, 80 years; males, 52%; main locations: digestive (36.3%), breast (16%), and urinary tract (14.8%); metastases, 43.5%). Area under the receiver operating characteristic curve (AUROC) estimates were compared between both tools against: (1) the detection of ≥1 or (2) ≥2 GA impairments, (3) the prescription of ≥1 geriatric intervention and the identification of an unfit profile according to (4) a latent class typology, expert-based classifications from (5) Balducci, (6) the International Society of Geriatric Oncology task force (SIOG), or using (7) a GA frailty index according to the Rockwood accumulation of deficits principle. AUROC values were ≥0.80 for both tools under all tested definitions. They were statistically significantly higher for the modified G8 for six reference standards: ≥1 GA impairment (0.93 vs. 0.89), ≥2 GA impairments (0.90 vs. 0.87), ≥1 geriatric intervention (0.85 vs. 0.81), unfit according to Balducci (0.86 vs. 0.80) and SIOG classifications (0.88 vs. 0.83), and according to the GA frailty index (0.86 vs. 0.84). Our findings demonstrate the robustness of both screening tools against different reference standards, with evidence of better diagnostic performance of the modified G8.

## 1. Introduction

In order to detect health problems in older patients with cancer and accordingly tailor treatment decisions, multidimensional geriatric assessment (GA) is recommended [1]. Because GA is time-consuming and requires specific expertise for its conduction, screening tools have been developed to help identify potentially frail patients warranting complete GA, following a two-step approach to be used particularly in clinical settings where performing GA is not feasible for all older patients. However, we have no unique definition of what this population constitutes, and what the reference standard should be. Although two main approaches have been proposed to define frailty, namely, the cumulative deficit model described by Rockwood et al. [2], and the physical phenotype developed by Fried [3], there is currently no consensus and no broadly accepted standard for measuring frailty in older cancer patients. Several classifications, usually based on clinical expertise and professional consensus, were used, but their concordance was variable, with different patients identified as frail depending on the used criteria [4].

In the geriatric oncology setting, a pragmatic definition based on ≥1 abnormal GA test assessing important domains at the GA (i.e., functional status, comorbidity, cognition, mood, nutrition) has mostly been used for developing and validating screening instruments [5,6,7], but this approach is hampered by a lack of standardization of GA components across studies. This definition also does not capture important aspects of the reality of clinical practice in geriatric oncology, such as actual treatment decisions based on GA findings, expert-based clinical classifications, and/or broader approaches to frailty.

The G8 [5] and modified G8 [6] screening tools were specifically developed for older patients with cancer using the definition of ≥1 GA impairment as the reference standard.

The original G8 was compared with the GA in 16 studies [5,6,8,9,10,11,12,13,14,15,16,17,18,19,20,21] of older patients with cancer; 7 studies used a cutoff for impairment of ≥1 GA deficiencies, reporting sensitivity ranging from 65% to 90% and mean specificity of 55% (range 3% to 100%). Twelve studies reported results using a cutoff for impairment of ≥2 GA deficiencies, with sensitivity ranging from 38% to 97% and specificity from 29% to 79%. Another study [16] of patients with hematologic disorders used Fried’s criteria [3] to assess the performance of the G8, reporting results of similar magnitude, with sensitivity of 82% and specificity of 51%. Outside the oncological setting, only few studies have evaluated screening tools against definitions other than the GA [22,23]. To our knowledge, no other reference standard has been tested with the G8. With the aim of improving the performance of the original G8, which is among the most sensitive tools but lacks specificity, the modified G8 was developed, achieving both appropriate sensitivity and specificity for predicting an abnormal GA. No other study has reported on the diagnostic performance of the modified G8 using reference standards other than an abnormal GA result. We, therefore, aimed to measure and compare the diagnostic performances of the original G8 versus the modified G8 using six other classifications evocative of a state of frailty.

## 2. Materials and Methods

### 2.1. Study Design and Patients

We studied patients recruited between January 2007 and June 2015 from the ELCAPA prospective cohort study, for whom complete data on the six reference standards and screening G8 scores were available (*n* = 1136, Figure 1). Consecutively enrolled patients ≥70 years old with a diagnosis of solid cancer or hematologic malignancy were referred for GA to one of ten geriatric oncology clinics in teaching hospitals in the Paris urban area. The study was approved by the institutional review board of the Henri-Mondor Teaching Hospital, Creteil, France, and each patient provided written informed consent before inclusion. All research was performed in accordance with relevant guidelines and regulations. The survey is registered at ClinicalTrials.gov (NCT02884375; accessed on 3 January 2022).

### 2.2. Reference-Standard Definitions

The term “reference standard” is used to describe the best available method for establishing the presence or absence of the condition of interest [24], and thus constitutes the ultimate measure for comparing new diagnostic or screening tests in testing accuracy studies. However, this situation assumes that an established reference standard is available and has perfect accuracy, which is not always the case. Reference-standard tests for many diseases may be difficult to implement because of their invasiveness, or may lack 100% accuracy or a clear cutoff value for the reference standard. In other cases, no unequivocal definition is available for the target condition, which prevents the characterization of a clear and definite reference standard.

In accordance with our research objectives, a wide spectrum of definitions was thus considered to check the ability of screening tools to identify a state evocative of frailty. The following reference-standard definitions evocative of a state of frailty were tested: (1) detection of ≥1 or (2) ≥2 impaired components of the GA, (3) prescription of ≥1 intervention by the geriatrician and identification of an unfit profile as defined by (4) a latent class typology (LCT) approach [25], (5) expert-based classifications from Balducci [26] and (6) the International Society of Geriatric Oncology task force (SIOG) classification [27], or using (7) a GA frailty index according to Rockwood accumulation of deficits principles [2].

#### 2.2.1. Geriatric Assessment

The GA included a variety of domains covering functional status, mobility, nutrition, cognition, mood, and comorbidities used in the development of the modified G8 screening tool [6] and in accordance with international recommendations [28]. Domains were evaluated by the following validated tests: Activities of Daily Living (abnormal: ADL ≤ 5/6), Instrumental Activities of Daily Living (abnormal: IADL ≤ 7/8), Timed Get Up-and-Go test (abnormal: TUG > 20 s), Mini Mental State Examination (abnormal: MMSE ≤ 23/30), mini Geriatric Depression Scale (abnormal: mini GDS ≥ 1/4), Mini Nutritional Assessment (abnormal: MNA ≤ 23.5/30), and Cumulative Illness Rating Scale for Geriatrics (CIRS-G; abnormal if ≥1 comorbidity grade 3 or 4). Considered thresholds were ≥1 and ≥2 impaired components.

#### 2.2.2. Geriatric Interventions

For each patient, proposed geriatric interventions after GA were documented. After internal review by two expert geriatricians (ML, PC) and for the present analysis, the final recommendation of the geriatrician for adapting the anticancer treatment was considered, as well as four domains covering clinically relevant deficiencies that may warrant further geriatric interventions: nutritional, home, neuropsychological, and social support. A consideration of ≥1 of these interventions prescribed by the geriatrician was defined as the reference standard.

#### 2.2.3. Frailty Classifications

Four classifications were considered to approach the non standardized definition of frailty: the Balducci and SIOG classifications, the LCT and a GA-derived frailty index, using the “unfit” profiles as reference standards.

Details regarding specific indicators and measures considered to classify patients as fit or unfit (regrouping the categories of vulnerable or frail or too sick) are given in Appendix A.

According to the Balducci classification derived from the criteria described by Balducci et al. [26,29], and as implemented in Ferrat et al. [4], fit patients were defined as functionally independent (no dependence in ADL and IADL), without serious comorbidity (retaining CIRS-G grade 0, 1, or 2 for the present analysis) and without geriatric syndromes, and unfit patients as dependent in one or more ADL (≤5/6) and/or one or more IADL (≤7/8) and/or with one or more severe comorbidities (retaining CIRS-G grade 3 or 4 for the present analysis) and one or more geriatric syndromes (Appendix A). Similarly to Ferrat et al. [4], considered geriatric syndromes included dementia, delirium, depression, urinary and/or fecal incontinence, and falls (≥1 fall in the last 6 months); three geriatric syndromes also used in Balducci, namely, osteoporosis, neglect and abuse, and failure to thrive, were not available in our database and were thus disregarded.

According to the SIOG classification [27], fit patients were defined as having no serious comorbidity (CIRS-G grade 0, 1 or 2), functionally independent (no dependence in IADL and ADL), and not malnourished and unfit patients as dependent in one or more ADL (≤5/6) or IADL (2 categorizations considered to define impairment: ≤7/8 for all patients; ≤7/8 for women and ≤3/4 for men) and/or with one or more severe comorbidities (CISR-G grade 3 or 4) and/or malnutrition (Appendix A). The original definition for malnutrition was not available in our database, so we used the following substitute, according to French guidelines [30]: ≥5% of weight loss in the last month and/or ≥10% within the last 6 months instead of ≥5% during the previous 3 months.

Additionally, we considered an LCT developed in a population of older patients with cancer, combining components of the GA [25]. Scoring equations were based on a set of indicators and covariates (Appendix A) yielding posterior class membership probabilities for each patient. A patient was classified as “fit” if the probability of membership in class 1 (relatively healthy) was ≥50% and unfit if the probability was <50%.

Lastly, a frailty index was constructed according to the cumulative deficit model developed by Rockwood et al. [2] and following recommendations for constructing a frailty index from Searle et al. [31]. A frailty index was derived from GA findings, considering 52 health deficits to be combined into a global index ranging from 0 to 1, where 0 corresponds to no deficit being present, and 1 to all 52 deficits being present (Appendix A). A cutoff of 0.3 was used to define fit (<0.3) and unfit (≥0.3) patients. While a lower cutoff of 0.2 was previously suggested by Searle et al., using this threshold would have defined 99% of our study population as unfit, preventing the conduct of robust analyses.

### 2.3. Screening Tools

The G8 screening tool includes 8 items (Appendix A). Total scores range from 0 to 17, a cutoff score of ≤14 defined as abnormal [5]. The modified G8 tool includes 6 items (Appendix A). Total scores range from 0 to 35, and a cutoff score of ≥6 was defined as abnormal [6].

### 2.4. Statistical Analysis

Sample size calculation was based on our preliminary work [7] evaluating the performance of the G8 and modified G8 in identifying older cancer patients likely to have an abnormal GA, estimating areas under the receiver operating characteristic curve (AUROC) at 86.5% and 91.6%, respectively. On the basis of a comparison of AUROC between the two instruments at two-sided 5% alpha risk, and considering an expected frailty prevalence of 90%, we calculated that the inclusion of at least 838 patients would yield a statistical power of 80% to identify a minimal difference in AUROC of 4% [32,33].

The study population was described in terms of clinical and demographic characteristics, and GA results. Univariate logistic regression analyses were used to assess the associations between the reference standards and both screening tools, estimating unadjusted odds ratios (ORs) and 95% confidence intervals (CIs). We tested for the equality of the regression coefficients for both tools. AUROC estimates were calculated to compare the diagnostic performance of both screening tools against the reference standards, with 95% CIs estimated. We tested for the equality of the AUROC values with an algorithm suggested by DeLong and Clarke-Pearson [34] for comparing both tools. We additionally investigated whether a different cutoff value provided a better discriminative performance for each reference standard. Sensitivities and specificities were calculated for optimal cutoff values (those prioritizing sensitivity), along with their 95% CIs, and were compared by McNemar’s chi-squared test. Positive predictive values (PPV), negative predictive values (NPV), positive likelihood ratios (LR+), and negative likelihood ratios (LR−) were additionally calculated. All tests were two-tailed, and the significance threshold was *P* < 0.05. All analyses used Stata v13 (StataCorp, College Station, TX, USA).

## 3. Results

### 3.1. Patient Characteristics and Geriatric Interventions

Main patient characteristics and GA results are in Table 1. Median age was 80 years (interquartile range (IQR) 76–85). The most frequent cancers were those of the digestive system (36.3%), followed by breast cancer (16%), and urinary tract cancer (14.8%), with almost half of the patients presenting metastasis (43.5%). A loss of functional capacities was common, with 31.6% and 58.5% of patients having at least one impairment in Activities of Daily Living (ADL) and Instrumental ADL (IADL), respectively. An impaired Mini Nutritional Assessment (MNA) was identified in 64.3% of patients. The burden of comorbidities was high, with 63.1% of patients having at least one comorbidity of severity grade 3 or 4 according to the Cumulative Illness Rating Scale for Geriatrics (CIRS-G) criteria.

A median of 3 interventions (IQR 2–5) were proposed for each patient. The most frequent intervention concerned nutritional support (74.5%) (Table 2). Physiotherapy and social support were proposed for 63.8% and 63.5% of patients, respectively. Overall, at least one intervention was proposed by the geriatrician for 91% of patients (N = 1032).

### 3.2. Prevalence of Unfit Patients by Reference Standard

The proportion of patients classified as unfit according to the reference standards varied (Table 3 and Table 4): 91.9% (GA: ≥1 impairment), 76.9% (GA: ≥2 impairments), 83.2% (International Society of Geriatric Oncology task force classification-SIOG), 89.4% (Balducci classification), 79.5% (Latent Class Typology-LCT) and 88.8% (GA frailty index).

### 3.3. Predictive Performance of Screening Tools by Reference Standard

On univariate logistic regression analyses (Table 3), abnormal G8 scores were significantly associated with all reference standards, regardless of definition. Nutritional support had the strongest association among types of interventions (G8: OR 8.6, 95% CI 5.9–12.6; modified G8: OR 9.1, 95% CI 6.3–13).

Area under the receiver operating characteristic curve (AUROC) values were ≥0.80 for both tools and all tested definitions (Figure 2). It was significantly higher for the modified G8 for six of the seven tested reference standards: ≥1 GA impairment (modified G8: 0.93 (95% CI 0.91–0.95) vs. original G8: 0.90 (0.87–0.92); *P* = 0.0029), ≥2 GA impairments (modified G8: 0.90 (0.88–0.92) vs. original G8: 0.87 (0.88–0.92); *P* = 0.0006), ≥1 geriatric intervention prescribed (modified G8: 0.85 (0.81–0.89) vs. original G8: 0.81 (0.77–0.86); *P* = 0.0056), unfit according to Balducci (modified G8: 0.86 (0.83–0.89) vs. original G8: 0.80 (0.76–0.83); *P* < 0.00001) and SIOG classifications (modified G8: 0.88 (0.86–0.91) vs. original G8: 0.83 (0.81–0.86); *P* < 0.00001) and GA frailty index (modified G8: 0.86 (0.83–0.90) vs. original G8: 0.84 (0.80–0.87); *P* = 0.029).

Table 4 details the diagnostic performance of each tool for the seven tested reference standards. Sensitivities based on optimal cutoffs ranged from 82% (original G8) and 84% (modified G8) for the Balducci classification, to 91% (both tools) for ≥1 GA impairment, with significant differences found for ≥2 GA impairments, Balducci and SIOG classifications, and GA frailty index in favor of the modified G8. Most specificities were higher for the modified G8, ranging from 41% (≥1 geriatric intervention prescribed) to 63% (GA frailty index) for the original G8 and from 56% (≥1 geriatric intervention prescribed) to 75% (≥1 GA impairment) for the modified G8. With the exception of the LCT, positive predictive values (PPVs) were higher, and negative predictive values (NPVs) were lower for the modified G8 compared to the original G8, with better LRs for the modified G8.

When considering the IADL four items for men and eight items for women in the definition of the SIOG classification (N = 1102), results were similar to those for the eight-item IADL for all patients. Sensitivities and specificities were 86.7% and 59.9% for the original G8, and 89.6% and 67.8% for the modified G8. AUROC values were significantly higher for the modified G8 than those of the original G8: 0.90 (95% CI 0.87–0.92) vs. 0.85 (0.82–0.87); *p* < 0.00001.

## 4. Discussion

In the present study of older cancer patients, we assessed the diagnostic performance of the original and modified G8 tools against different reference standards to evaluate their robustness. We tested seven reference standards evocative of a geriatric risk profile. Regardless of the tested reference standard, both tools demonstrated high predictive value and performance robustness to detect various definitions evocative of frailty. Statistically significant differences in AUROCs favored the modified G8 over the original G8 for ≥1 and ≥2 GA impairments, a geriatric intervention, the Balducci and SIOG classifications and the GA frailty index, demonstrating better screening performance of the modified G8 (six of the seven tested reference standards). AUROC findings for both the G8 and modified G8 predicted subsequent prescription of geriatric interventions for relevant clinical domains. This finding is of particular clinical relevance because it relates directly to the main objective of the screening tools to identify patients who would benefit from a complete GA. Beyond conceptual pitfalls to define frailty, this finding further supports the pragmatic aim of the G8 instruments to adequately detect patients with potential deficits warranting interventions and the optimization of treatments.

The G8 and modified G8 screening tools were originally developed to identify patients with ≥1 impairment in a multidimensional GA, proposed by the SIOG [1] as the reference standard for evaluating older cancer patients to determine the optimal oncologic treatment. However, a standardized definition of GA and, more importantly, an abnormal GA is lacking. Indeed, the definition of an abnormal GA varies greatly across studies, which may use a different number of components, and different scales and thresholds for defining impairment, hence limiting the comparability of study results [7]. Furthermore, this pragmatic definition most often used in the literature does not correspond well to the reality of clinical practice of geriatricians and oncologists, having limited applicability and representing a problem for implementation in routine clinical care.

Other frailty classifications have been developed to help physicians select the best cancer treatment and guide geriatric interventions. In a recent study, the prognostic value of three classifications (Balducci, SIOG, and the LCT) was assessed and found to be good for 1-year mortality and 6-month unscheduled hospitalizations in older cancer patients [4], supporting their use to stratify older cancer patients according to their health status for clinical decision making, and also as a candidate reference definition for screening-test accuracy studies because of their predictive value for patient outcomes [35]. For example, some frailty criteria have been used to help evaluate the toxic effects of treatments.

On the basis of our results, the modified G8 seemed to be an appropriate tool to identify several profiles suggesting frailty, regardless of the definition that has been continuously debated over the past decades.

The present study is the first to thoroughly examine the variability of the diagnostic performance of screening tools for frailty in older patients with cancer under multiple clinically relevant reference standards or definitions. Adding to the previously reported high prognostic value of the two instruments [36], our findings reinforce the clinical utility of the G8 tools in daily geriatric oncologic practice.

Our study has some limitations. First, patients from our study population were referred to a geriatrician for a GA with varying and sometimes limited rates of tumor locations (e.g., 7%, 11%, and 16% for hematological, prostate, and breast cancers, respectively); our results may thus not completely reflect the real-life population of older patients with cancer. Second, data were missing for some key variables to compute G8 scores and/or reference standards, although missing rates per variable were overall low (median 7%, range 0–17.6%). Patients were also excluded from analysis when data on any of the six reference standards were not available to allow for the direct comparison of the screening tools using a common population. We found no statistically significant difference between included and excluded patients in demographic and clinical characteristics. In addition, there would have been interest in assessing other reference standards, such as the Fried phenotype [3], a well-established instrument developed for the general geriatric population, but not specifically for older patients with cancer.

Further studies are necessary to corroborate our findings and to evaluate the predictive value of both tools for other relevant outcomes, namely, functional decline, treatment-related toxicity, and quality of life. In particular, it would be of special interest to determine if geriatric management integrating the G8 or the modified G8 ultimately improves patient health outcomes.

## 5. Conclusions

Our findings demonstrate the robustness of the original and modified G8 against different reference standards, further supporting the clinical value of these instruments for detecting older patients with cancer who warrant a complete GA. The modified G8 demonstrated better diagnostic performance than that of the original G8 for detecting a variety of health profiles evocative of frailty. Our findings may offer a practical response for daily practice with an instrument able to detect any potential risk problem regardless of definition.

## Figures and Tables

**Figure 1 cancers-14-00244-f001:**
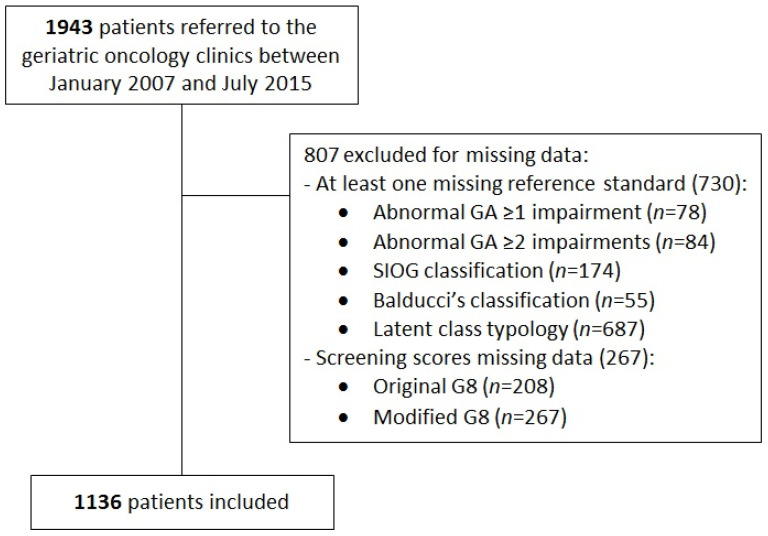
Flow diagram of patients in the study. GA, geriatric assessment; SIOG, International Society of Geriatric Oncology.

**Figure 2 cancers-14-00244-f002:**
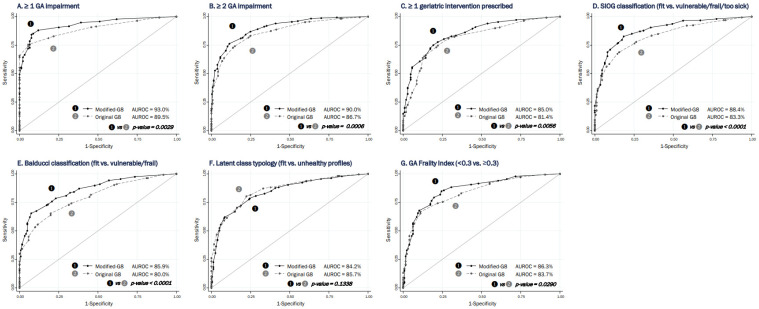
Receiver operating characteristic (ROC) curves for predicting different reference standard definitions: original vs. modified G8 questionnaire. (**A**) ≥1 GA impairment; (**B**) ≥2 GA impairments; (**C**) ≥1 geriatric intervention prescribed; (**D**) SIOG classification (fit vs. vulnerable/frail/too sick); (**E**) Balducci classification (fit vs. vulnerable/frail); (**F**) latent class typology (fit vs. “unhealthy profiles”); (**G**) GA frailty index. AUROC: area under the ROC curve; GA: geriatric assessment; SIOG, International Society of Geriatric Oncology.

**Table 1 cancers-14-00244-t001:** Patient baseline characteristics (N = 1136).

Characteristics	N	%
Age, years, median (IQR)	80 (76–85)	
No. of medications/day, median (IQR)	6 (4–8)	
Outpatient	412	36.3
Male gender	587	51.7
Cancer type		
Colorectal	201	17.7
Liver or upper gastrointestinal tract	211	18.6
Urinary tract	168	14.8
Prostate	127	11.2
Hematological	84	7.4
Breast	182	16.0
Others ^a^	163	14.3
Metastasis	494	43.5
Inappropriate social environment ^b^	177	15.6
Functional impairment		
ADL ≤ 5	359	31.6
IADL ≤ 7	665	58.5
ECOG-PS		
0: Fully active	205	18.0
1: Restricted activity	342	30.1
≥2: Unable to carry out work activities/confined to bed >50% or disabled	589	51.9
Cognitive impairment (MMSE ≤ 23)	285	25.1
Depressive disorder		
Mini-GDS ≥ 1	379	33.4
DSM IV criteria	364	32.0
Malnutrition		
Impaired MNA ≤ 23.5	730	64.3
HAS criteria ^c^	776	68.3
At risk of malnutrition or severe malnutrition ^d^	317	27.9
Comorbidities		
CIRS-G (≥1, grade 3/4)	717	63.1
Mobility		
TGUG ≥ 20 s	434	38.2
Falls during the previous 6 months	365	32.1

Abbreviations: IQR, interquartile range; ECOG-PS, Eastern Cooperative Oncology Group-Performance Status; ADL, Activities of Daily Living; IADL, Instrumental ADL; MMSE, Mini-Mental State Evaluation; GDS, Geriatric Depression Scale; DSM, Diagnostic and Statistical Manual of Mental Disorders; MNA, Mini Nutritional Assessment; HAS, French National Authority for Health; CIRS-G, Cumulative Illness Rating Scale for Geriatrics; TGUG, Timed Get Up-and-Go test; SIOG, International Society of Geriatric Oncology. ^a^ Lung (N = 44), skin (N = 32), unknown primary origin (N = 30), sarcoma (N = 15), gynecologic (N = 14), brain (N = 11), head and neck (N = 5), thyroid (N = 3), others (N = 9). ^b^ Defined as absence of a primary caregiver or adequate support at home or a strong circle of family and friends able to meet the needs of the patient at time of evaluation. ^c^ One or more of: at least 10% weight loss in 6 months or 5% in 1 month and/or body mass index < 21 kg/m^2^ and/or MNA score < 17/30 and/or serum albumin level < 35 g/L. ^d^ Weight loss ≥ 10% in the last 6 months and/or ≥5% in the last month.

**Table 2 cancers-14-00244-t002:** Geriatric interventions for overall patient management.

Geriatric Interventions	N	%
≥1 Nutritional support	846	74.5
Dietary advice	744	65.5
Nutritional supplements	353	31.1
≥1 Home support	741	65.2
Physiotherapy	725	63.8
Nursing	77	6.8
≥1 Social support	721	63.5
Social care	665	58.5
Personal assistance	315	27.7
Personal care allowance (APA)	224	19.7
≥1 Neuropsychological support	438	38.6
Psychological care	431	37.9
Psychiatric care	46	4.1
Adaptation of anticancer treatment	263	23.2
≥1 interventions prescribed	1032	90.9

**Table 3 cancers-14-00244-t003:** Association between reference standards and both screening tools (original G8 and modified G8), N = 1136.

Reference Standards	Screening Tools	Normal Scores ^a^	Abnormal Scores ^b^		
		N (%)	N (%)	OR (95% CI)	*p* Values
≥1 GA impairment (N = 1044)	Original G8	97 (9.3%)	947 (90.7%)	11.62 (7.33–18.42)	<0.0001
	Modified G8	98 (6.4%)	946 (90.6%)	28.96 (17.29–48.50)	<0.0001
≥2 GA impairments (N = 874)	Original G8	42 (4.8%)	832 (95.2%)	13.25 (8.91–19.69)	<0.0001
	Modified G8	40 (4.6%)	834 (95.4%)	19.61 (13.16–29.24)	<0.0001
≥1 Interventions prescribed (N = 1032)	Original G8	104 (10.1%)	928 (89.9%)	6.29 (4.05–9.76)	<0.0001
	Modified G8	109 (10.6%)	923 (89.4%)	10.68 (6.91–16.49)	<0.0001
≥1 Nutritional support	Original G8	48 (5.7%)	798 (94.3%)	8.62 (5.90–12.59)	<0.0001
	Modified G8	55 (6.5%)	791 (93.5%)	9.05 (6.30–12.99)	<0.0001
≥1 Home support	Original G8	65 (8.8%)	676 (91.2%)	2.72 (1.92–3.87)	<0.0001
	Modified G8	53 (7.2%)	688 (92.8%)	5.27 (3.70–7.50)	<0.0001
≥1 Social support	Original G8	64 (8.9%)	657 (91.1%)	2.57 (1.81–3.65)	<0.0001
	Modified G8	74 (10.3%)	647 (89.7%)	2.53 (1.81–3.52)	<0.0001
≥1 Neuropsychological support	Original G8	26 (5.9%)	412 (94.1%)	3.32 (2.14–5.17)	<0.0001
	Modified G8	33 (7.5%)	405 (92.5%)	2.92 (1.95–4.36)	<0.0001
Treatment adaptation	Original G8	27 (10.3%)	236 (89.7%)	1.39 (0.89–2.17)	0.142
	Modified G8	25 (9.5%)	238 (90.5%)	1.84 (1.18–2.90)	0.007
SIOG classification (unfit ^c^:N = 945)	Original G8	74 (7.8%)	871 (92.2%)	8.31 (5.66–12.20)	<0.0001
	Modified G8	69 (7.3%)	876 (92.7%)	14.92 (10.14–21.94)	<0.0001
Balducci classification (unfit ^d^:N = 1015)	Original G8	99 (9.8)	916 (90.2)	6.08 (4.00–9.25)	<0.0001
	Modified G8	104 (10.3)	911 (89.7)	9.51 (6.31–14.34)	<0.0001
Latent class typology (unfit ^e^:N = 903)	Original G8	55 (6.1%)	848 (93.9%)	10.06 (6.89–14.69)	<0.0001
	Modified G8	69 (7.6%)	834 (92.4%)	8.77 (6.14–12.55)	<0.0001
GA frailty index (unfit ^f^:N = 1009)	Original G8	90 (8.9%)	919 (91.1%)	8.31 (5.51–12.54)	<0.0001
	Modified G8	95 (9.4%)	914 (90.6%)	12.59 (8.36–18.97)	<0.0001

GA, geriatric assessment; OR, odds ratio; 95% CI, 95% confidence interval. ^a^ G8: >14 points; modified G8: <6 points. ^b^ G8: ≤14 points; modified G8: ≥6 points. ^c^ Vulnerable or frail or too sick. ^d^ Latent classes 2 to 4: Malnourished or cognitive/mood impaired or globally impaired. ^e^ Vulnerable or frail. ^f^ GA frailty index ≥ 0.3.

**Table 4 cancers-14-00244-t004:** Diagnostic performance of G8 and modified G8 according to reference standards.

Reference Standards	Prevalence	Screening Tools	Cutoffs ^a^	Sensitivity (95% CI)	*p* Value ^b^	Specificity (95% CI)	*p* Value ^b^	PPV	NPV	LR+	LR−
GA ≥1 impairment	91.9%	Original G8	≤14	90.7% (88.8–92.4)	0.921	54.3% (43.6–64.8)	**0.001**	95.8%	34.0%	1.99	0.17
		Modified G8	≥6	90.6% (88.7–92.3)		75.0% (64.9–83.4)		97.6%	41.3%	3.62	0.13
GA ≥2 impairments	76.9%	Original G8	≤13.5	88.7% (86.4–90.7)	**0.002**	58.4% (52.2–64.4)	**0.048**	87.8%	60.7%	2.14	0.19
		Modified G8	≥8	91.9% (89.9–93.6)		64.8% (58.6–70.5)		89.7%	70.4%	2.61	0.13
≥1 Geriatric intervention	90.9%	Original G8	≤14	89.9% (87.9–91.7)	0.629	41.4% (31.8–51.4)	**0.005**	93.8%	29.3%	1.53	0.24
		Modified G8	≥6	89.4% (87.4–91.2)		55.8% (45.7–65.5)		95.3%	34.7%	2.02	0.19
SIOG classification ^c^	83.2%	Original G8	≤13.5	84.8% (82.3–87.0)	**0.006**	59.8% (52.1–67.1)	**0.016**	92.0%	41.9%	2.11	0.25
		Modified G8	≥8	87.7% (85.5–89.7)		69.0% (61.5–75.7)		93.9%	50.8%	2.83	0.18
Balducci classification ^d^	89.4%	Original G8	≤13.5	81.7% (79.2–84.0)	**0.023**	54.5% (45.2–63.6)	**0.009**	93.8%	26.2%	1.80	0.34
		Modified G8	≥8	84.1% (81.7–86.3)		65.3% (56.1–73.7)		95.3%	32.9%	2.42	0.24
Latent class typology ^e^	79.5%	Original G8	≤13.5	88.2% (85.9–90.2)	0.999	62.2% (55.7–68.5)	0.103	90.0%	57.5%	2.33	0.19
		Modified G8	≥8	88.2% (85.9–90.2)		57.1% (50.5–63.5)		88.8%	57.1%	2.05	0.21
GA frailty index ^f^	88.8%	Original G8	≤13.5	83.0% (80.5–85.2)	**0.020**	63.0% (54.0–71.4)	**0.002**	94.7%	31.7%	2.24	0.27
		Modified G8	≥8	85.5% (83.2–87.6)		74.0% (65.5–81.4)		96.3%	39.2%	3.29	0.19

95% CI, 95% confidence interval; PPV, positive predictive value; NPV, negative predictive value; LR+, positive likelihood ratio; LR−, negative likelihood ratio; GA, geriatric assessment; SIOG, International Society of Geriatric Oncology. ^a^ Official cutoff for GA ≥1 impairment and best cutoff (prioritizing sensitivity) otherwise. ^b^ Original vs. modified G8 (McNemar’s chi-squared test). ^c^ Fit vs. unfit (vulnerable or frail or too sick) ^d^ Fit vs. unfit (vulnerable or frail). ^e^ Fit vs. unfit (latent classes 2 to 4: malnourished or cognitive/mood impaired or globally impaired). ^f^ Fit vs. unfit (<0.3 vs. ≥0.3). Bolded *p*-values are statistically significant at the *p* < 0.05 level.

## Data Availability

Restrictions apply to the availability of these data. Data were obtained from the ELCAPA Study Group and are available from the corresponding author with the permission of the ELCAPA Study Group investigators.

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
