# Peer review of "Predicting Frailty and Geriatric Interventions in Older Cancer Patients: Performance of Two Screening Tools for Seven Frailty Definitions—ELCAPA Cohort"

_cancers, 2022, doi:10.3390/cancers14010244_

Round 1

Reviewer 1 Report

The authors have submitted an interesting study on the predictive ability of frailty of two screening tests. However, the manuscript presents some deficiencies that must be correctly addressed before a possible publication.

Abstract section.
The reviewer suggests that the authors include a brief description of the sample profile.
Introduction section The reviewer finds this section confusing. In the first place, the definition of "reference standard", if its detail is precise, it would be more logical to include it in the methods section. In the second paragraph it is commented that the multidimensional geriatric assessment (GA) is time consuming. That's true, but it's no less true that neither the G8 nor the modified G8 were designed as replacements for the GA. The same happens with the third paragraph. If the objective is to know the predictive capacity of fragility of the G8 and the modified G8 by varying the reference-standards definitions, the logical thing would be to comment on the sensitivity or specificity problems of the G8 scale and that the objective is to analyze, as described later in methods , how modified G8 improves the prognostic ability of frailty screening.
Methods section
The reviewer advises including the calculation of the sample size and the power of the study.
Results section The reviewer advises the authors to remove the first three lines of the results section. The reviewer is under the impression that they have been added from the text of the journal's recommendations. Likewise, the reviewer congratulates the authors for the clarity of the results presentation.
Discussion section The reviewer considers that some of the data provided on the sensitivity and specificity of the oncogeriatric frailty screening scales should be part of the state of the art that is usually described in the introduction. Furthermore, the reviewer considers that the authors should detail the other biases present in the study in the limited subsection. Similarly, the conclusions section is too long and confusing. I agree with the authors about the good diagnostic capacity of the modified G8 scale, according to the results of the study. Nevertheless, it is not necessary to use so much space to specify the main findings of the study.

Reviewer 2 Report

This article is very interesting

The rate of people with breast cancer is low. This may be questioning the results. Indeed, during the construction of te G8 scale, it was noted a different sensivity and specificity for this cancer. This element should probably be added in the limitations.

Author Response

We thank the reviewer for her/his comments. We have added a sentence about the generalizability of the results in the limitations subsection of the Discussion, as follows:

First, patients from our study population were referred to a geriatrician for a GA, with varying and sometimes limited rates of tumor locations (e.g. 7%, 11% and 16% for hematological, prostate and breast cancers, respectively); our results may thus not completely reflect the real-life population of older patients with cancer.”

Round 2

Reviewer 1 Report

The authors have adequately addressed the comments made by the reviewer. For this reason, the reviewer wants to congratulate the authors, the reading of the current version of the manuscript is much more understandable for the reader.